# Analysis of the Manufacturing Variables of Binderless Panels Made of Leaves of Olive Tree (*Olea europaea* L.) Pruning Waste



Antonio Ferrandez-Garcia [1] [iD], Maria Teresa Ferrandez-Garcia [1] [iD], Teresa Garcia Ortuño [1], Francisco Mata-Cabrera [2] and Manuel Ferrandez-Villena [1,*] [iD]

1 Department Engineering, Universidad Miguel Hernández, 03300 Orihuela, Spain; antonio.ferrandezg@umh.es (A.F.-G.); mt.ferrandez@umh.es (M.T.F.-G.); tgarcia@umh.es (T.G.O.)
2 Department of Applied Mechanics and Project Engineering, Universidad Castilla-La Mancha, 13400 Almadén, Spain; francisco.mcabrera@uclm.es
* Correspondence: m.ferrandez@umh.es; Tel.: +34-966-749-716

**Abstract:** While the construction industry consumes more raw materials than any other industrial sector, agriculture generates a large amount of waste that is not managed properly. The olive industry produces more than 7.5 million tons of waste that could be recovered. This paper presents a new method to valorize the leaves of olive tree pruning waste consisting of the manufacture of ecologic boards without adhesives by hot pressing. In order to analyze their influence, three manufacturing variables were varied to obtain the boards: leaf type (shredded and whole leaves), temperature (130, 140 and 150 °C) and time (4, and 12 min). Twenty-four boards were made and were then tested for their mechanical, physical and thermal properties according to the EN standards. The boards showed good results of thickness swelling (TS), water absorption (WA) and of thermal conductivity and can be used as an alternative for manufacturing thermal insulation boards. With a smaller particle size of shredded leaves, longer pressing times and higher pressing temperatures, the mechanical behavior of the boards could improve. The olive leaves are a low-cost renewable resource, and manufacturing products with a long, useful life can be beneficial to the environment.

**Keywords:** agricultural residues; valorization; *Olea europaea* L.; fiberboard; particleboard

## 1. Introduction

Nowadays, there is great social concern about climate change, the environment, biodiversity and sustainable development. The European Union has set the goal to be the first climate-neutral continent by 2050. Both the European Parliament and the Council have agreed that the reduction of greenhouse gases will be 55% by 2030 compared to 1990 emissions.

While the construction industry consumes more raw materials than any other industrial sector [1], agriculture generates a large amount of waste that is not managed properly. The use of vegetable fiber-based materials in construction can alleviate this problem and improve the sustainability of the sector, due to the fact that they are easily recyclable and are not aggressive with the environment. The recovery of agricultural waste will also improve the reinforcement of European environmental policy, such as the Framework Directive on waste [2], the European Directive on industrial emissions [3] and the Directive on the landfill of waste [4].

In Spain, there are 2,623,721 hectares planted with olive trees, and it is the leading country in the production of olives and olive oil. This industry produces more than 7.5 million tons of waste that could be recovered [5]. Pruning produces leaves (approximately 25% wt), thin branches (approximately 50% wt) and thick branches or wood (approximately 25% wt), although the proportions may vary depending on the culture conditions, tree age, production and/or local pruning practices [6].

The use of olive pruning waste has been investigated in: basketry [7], animal feed [8], as a supplement in minced beef meet [9], in substrates for crops [10,11], activated carbons [12,13], bioremediation and bioabsorbents for soils [14–16], compost and biochar [17–21], cellulose nanofibers [22,23], Kraft pulp for paper [24], lignin evaluation [25,26], sugars and natural antioxidants [27,28], bioactive compounds [29,30] and polyphenols [31]. However, the majority of the studies of olive tree pruning are focused on the production of energy: biomass [32,33], pellets, briquettes and charcoal [34,35] and biofuel—both liquid and gaseous [36–40].

A different research line aims to valorize olive tree waste developing different construction materials: thermal and acoustic insulating panels made with olive tree pruning fibers and sodium silicate [41]; sound-absorbing materials with olive pruning waste and a chitosan binder [42]; bio-based plasters of olive fibers and a mixture of sand and clay [43]; ceramic lightweight bricks with three different olive tree residues: leaves, olive tree pruning and olive wood [44]; ceramic bricks with pine–olive pruning ash, olive stone ash and olive pomace ash [45]; the manufacture of cement with olive pomace and stone for replacing clinker [46]; and particle board with 80% olive stone and 20% polyester [47].

The present work presents for the first time a new method to valorize the leaves of olive tree pruning waste consisting of the manufacture of ecologic boards without adhesives. The aim is to analyze the variables of the manufacture process to develop a product that can be used in the building sector to counter the high consumption of materials of this industry by using an easily renewable resource.

## 2. Materials and Methods

### 2.1. Materials

The materials used to manufacture boards were leaves from olive tree pruning (*Olea europaea* L.) and water from the municipal network.

The leaves were obtained from pruning operations carried out by the Higher Technical College of Orihuela at Universidad Miguel Hernández, Elche. The pruning waste was left to dry outdoors for 3 months (Figure 1a). It was then classified in leaves and branches (Figure 1b). The approximate moisture content of the leaves was 7.4%. Half of the leaves were shredded in a laboratory-scale ring-knife chipper. Whole leaves were 40 to 60 mm in length, and shredded leaves had a particle size of 1 to 4 mm.

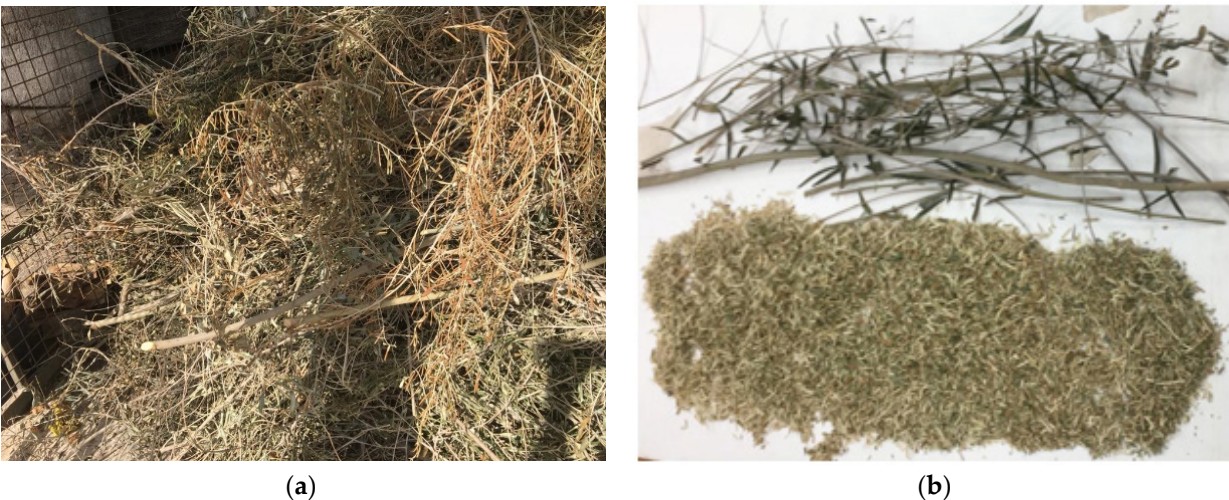

(**a**)             (**b**)

**Figure 1.** (**a**) Olive tree pruning waste; (**b**) leaves from a branch.

*2.2. Methods*

2.2.1. Board Manufacture

The method applied to manufacture the boards was the conventional dry process used in the industry. First the material was put into a mold of 400 × 600 mm, and then, a 3% wt of water was sprayed on the surface before placing it in the hot press (Figure 2a).

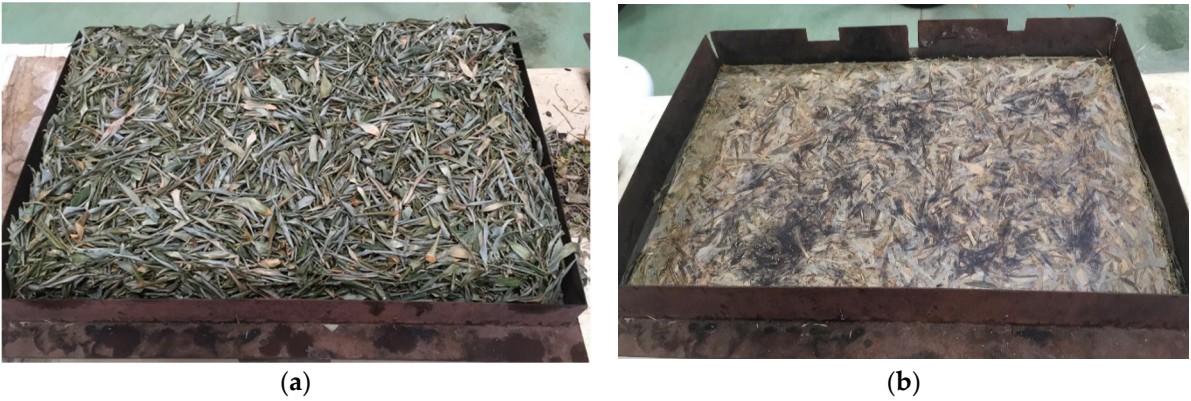

(**a**)      (**b**)

**Figure 2.** (**a**) Material in the mold; (**b**) panel manufactured in the hot press.

Previous experiences in the laboratory indicated that this was the most suitable amount of addition of water required for the self-bonding process. Contrary to the common 10% of added water in the literature [48], when more than 9% was added, the boards exploded in the press.

In order to analyze their influence, 3 manufacturing variables were varied to obtain the boards: leaf type (shredded and whole leaves), temperature (130, 140 and 150 °C) and time (4, and 12 min) while the pressure was maintained in 2.1 MPa (Figure 2b). Twenty-four boards were manufactured as shown in Table 1.

**Table 1.** Types of board manufactured.

| Type | Leaf Type | Temperature (°C) | Time (min) | Number |
|------|-----------|------------------|------------|--------|
| S1a | Shredded | 130 | 4 | 2 |
| S1b | Shredded | 130 | 12 | 2 |
| S2a | Shredded | 140 | 4 | 2 |
| S2b | Shredded | 140 | 12 | 2 |
| S3a | Shredded | 150 | 4 | 2 |
| S3b | Shredded | 150 | 12 | 2 |
| W1a | Whole | 130 | 4 | 2 |
| W1b | Whole | 130 | 12 | 2 |
| W2a | Whole | 140 | 4 | 2 |
| W2b | Whole | 140 | 12 | 2 |
| W3a | Whole | 150 | 4 | 2 |
| W3b | Whole | 150 | 12 | 2 |

The samples were cut to the appropriate dimensions (Figure 3), as indicated in the European Standards [49], in order to carry out the tests needed to characterize the mechanical, physical and thermal properties of each of the boards studied.

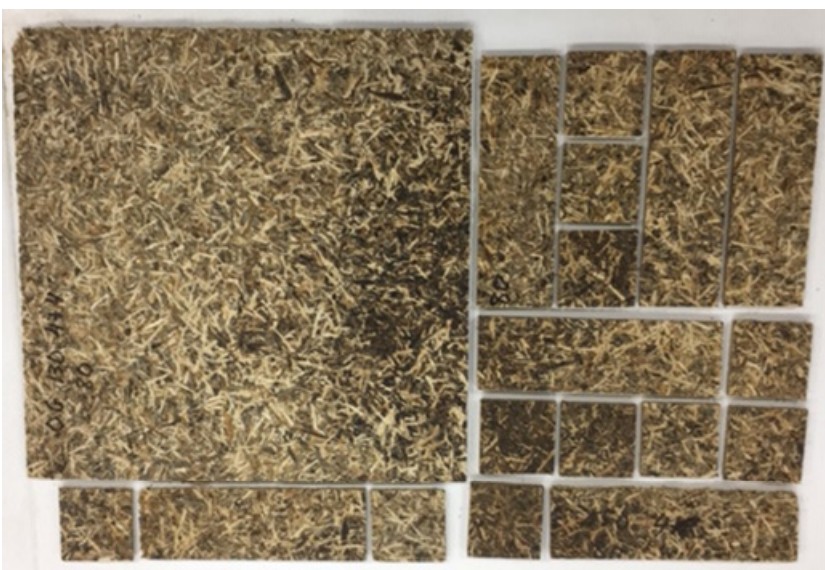

**Figure 3.** Specimens for the different tests.

2.2.2. Experimental Tests

The method that was followed was experimental. The tests were conducted in the Materials Strength Laboratory of the Higher Technical College of Orihuela at Universidad Miguel Hernández, Elche. The values were determined according to the European Standards established for wood particleboards [50].

After they were manufactured, density [51], thickness swelling (TS) and water absorption (WA) after 2 and 24 h immersed in water [52], internal bonding strength (IB) [53], modulus of elasticity (MOE) and modulus of rupture (MOR) [54] and thermal conductivity and resistance [55] were measured (Table 2). The boards were finally evaluated according to the European Standard [56].

**Table 2.** Characteristics of the tests performed.

| Test | N of Replicates (Per Panel) | Size of the Specimens | Equipment Used |
|---|---|---|---|
| Relative Humidity | 3 | 20 g | Model UM2000, Imal s.r.l. |
| Density | 6 | 50 × 50 mm | Model IB700, Imal s.r.l. |
| Thickness Swelling (TS) | 3 | 50 × 50 mm | Model 76-B0066/B Water Bath, Controls S.A. Model UM2000, Imal s.r.l. |
| Water Absorption (WA) | 3 | 50 × 50 mm | Model 76-B0066/B Water Bath, Controls S.A. Model UM2000, Imal s.r.l. |
| Modulus of Rupture (MOR) | 6 | 150 × 50 mm | Model UM2000, Imal s.r.l. |
| Modulus of Elasticity (MOE) | 6 | 150 × 50 mm | Model UM2000, Imal s.r.l. |
| Internal Bonding Strength (IB) | 3 | 50 × 50 mm | Model UM2000, Imal s.r.l. |
| Thermal Conductivity | 1 | 300 × 300 mm | NETZSCH Instruments Inc. |

The moisture content of the material was measured with a laboratory moisture meter (model UM2000, Imal S.R.L, Modena, Italy). For the panels, the water immersion test was carried out in a heated tank (Model 76-B0066/B Water Bath, Equipos de Ensayo Controls S.A., Toledo, Spain). The thermal conductivity and resistance tests were conducted with a heat flow meter (NETZSCH Instruments Inc., Burlington, MA, USA).

The mechanical tests and density were performed with the universal testing machine (model IB700, Imal, S.R.L., Modena, Italy), which complies with the velocity of 5 mm·min$^{-1}$ for the bending test and 2 mm·min$^{-1}$ for internal bonding strength.

For the statistical analyses, SPSS v. 28.0 software (IBM, Chicago, IL, USA) was used. An analysis of variance (ANOVA) and Pearson correlation were performed. The standard deviation was obtained for the mean values of the tests.

## 3. Results and Discussion

### 3.1. Physical Properties

The resulting boards were 5.95 ± 0.90 mm thick. Their average density could be classified as medium-high, which is in accordance with other binderless particleboards obtained by other authors. Boards with whole leaves had an average density of 887.8 kg/m³ and 936.3 kg/m³ when the leaves were shredded, as shown in Figure 4. It is possible that whole leaves were less compressed on the hot press. More time and temperature in the hot press resulted in higher density.

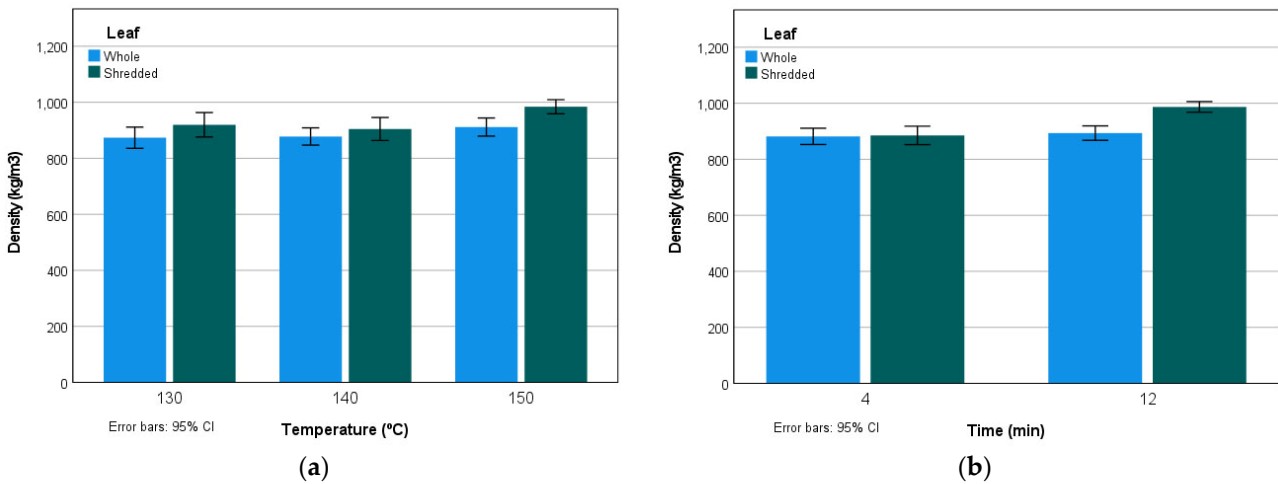

**Figure 4.** Density according to (**a**) temperature and (**b**) pressing time. CI: confidence interval.

Whole-leaf panels produced lower thickness swelling (TS) at 24-h (13.66 to 18.33%) than when shredded (from 19.04 to 39.35%) as shown in Figure 5. The amount of natural waxes in the whole leaves probably had an influence on this property, acting as a repellent. At a higher pressing temperature, the panels had more TS. Whereas pressing time did not seem to have any effect on this property in whole leaves, it reduced it on shredded leaves. P3 type boards (non-structural boards for use in a humid environment) [56] had a limit of 17% TS 24 h; therefore, some boards met this requirement.

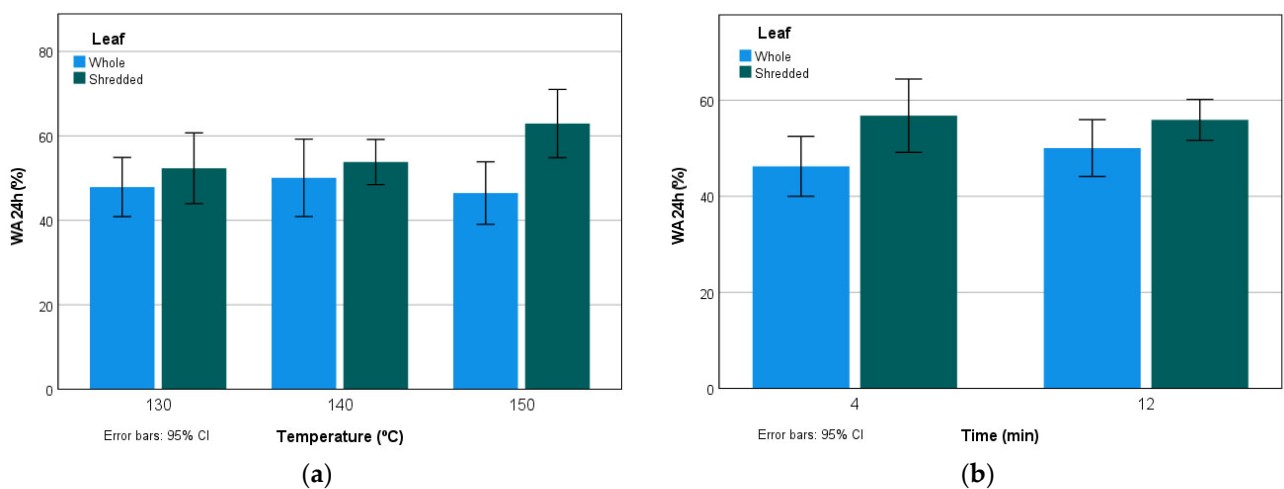

**Figure 5.** Thickness swelling (TS) after 24 h according to (**a**) temperature and (**b**) pressing time.

Figure 6 shows that WA after 24 h was higher when the leaves were shredded. Whole-leaf boards could absorb water without swelling as much as shredded leaves. It is possible that in shredded-leaf panels, pressing time influenced this property.

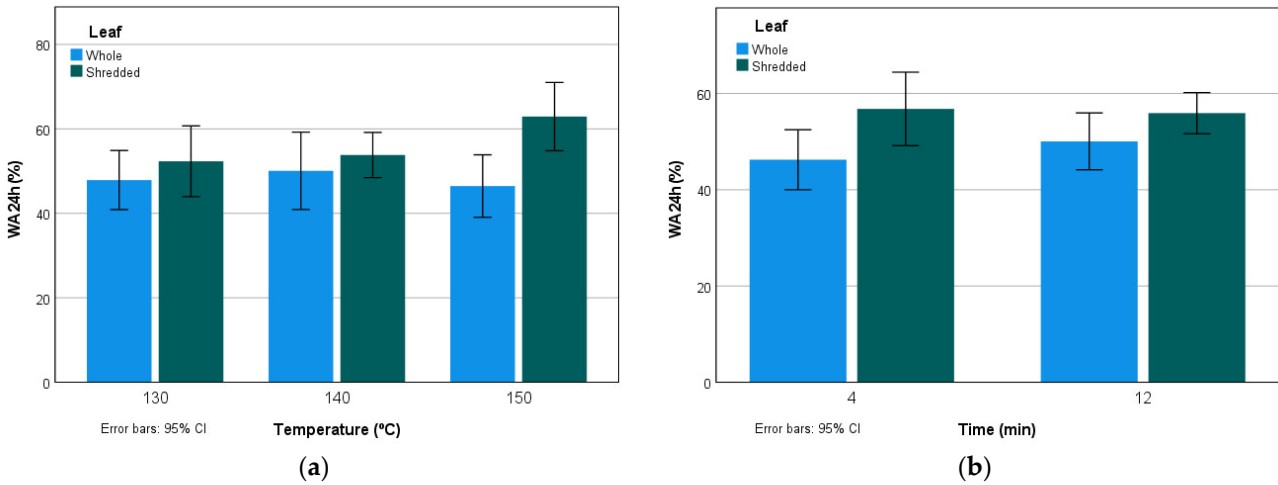

**Figure 6.** Water absorption (WA) after 24 h according to (**a**) temperature and (**b**) pressing time.

### 3.2. Mechanical Properties

The mean values of the modulus of rupture (MOR) are shown in Figure 7. Whole-leaf boards achieved an MOR of 5.02 N/mm$^2$ and shredded boards 5.67 N/mm$^2$. When manufactured with longer time and temperature in the hot press, it could be possible to strengthen the shredded panels.

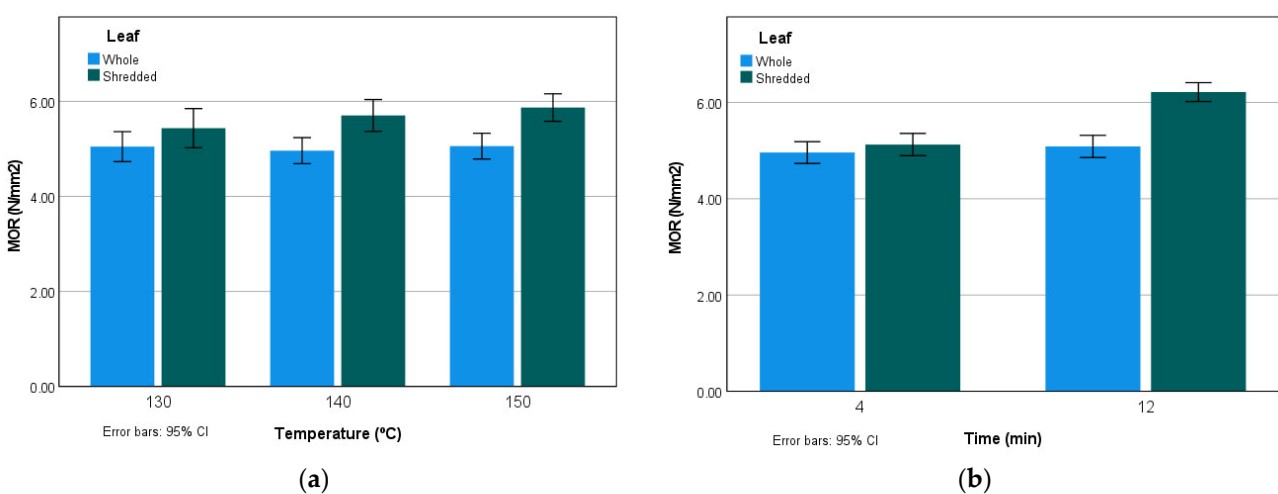

**Figure 7.** Modulus of rupture (MOR) according to (**a**) temperature and (**b**) pressing time.

As can be observed in Figure 8, the greater modulus of elasticity (MOE) was achieved when the boards were manufactured at 150 °C with shredded leaves. As in the previous property, with shredded leaves, more time and temperature in the hot press improved the results. Pressing temperature had no effects on whole leaves contrary to pressing time.

The internal bonding strength (IB) of the panels had a large variability of values, as shown in Figure 9. IB was higher in shredded-leaf boards with a mean value of 0.051 N/mm$^2$ than in whole leaves (0.012 N/mm$^2$). The longer the panels were kept in the hot press, the results in the shredded-leaf boards improved, but the deviations were very high with temperature. The IB of the whole leaves probably indicated that the self-bonding was only due to the waxes of the surface on the leaves and not to other mechanisms that may have been occurring when the leaves were shredded.

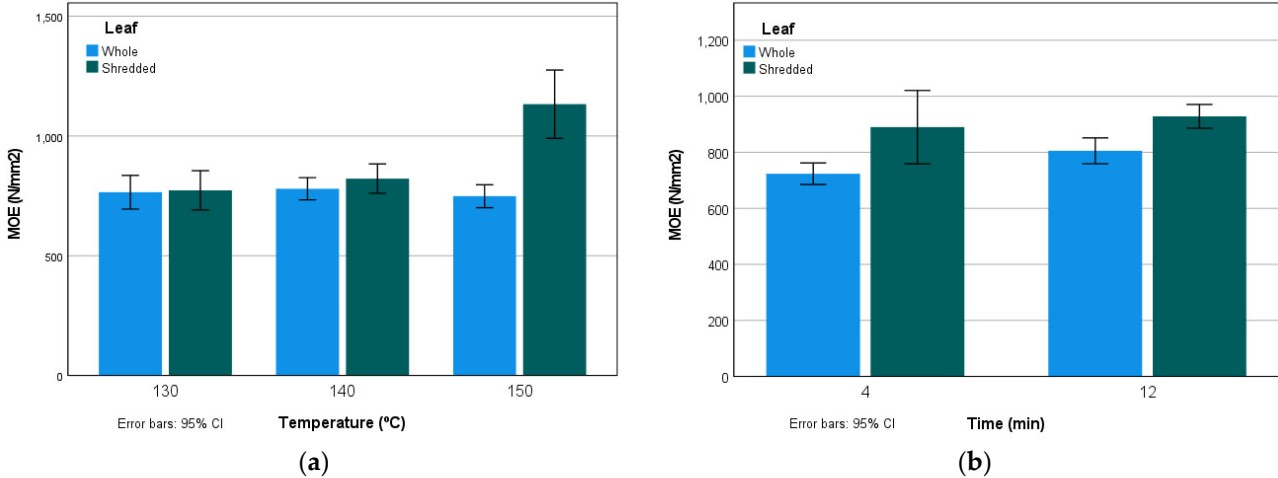

**Figure 8.** Modulus of elasticity (MOE) according to (**a**) temperature and (**b**) pressing time.

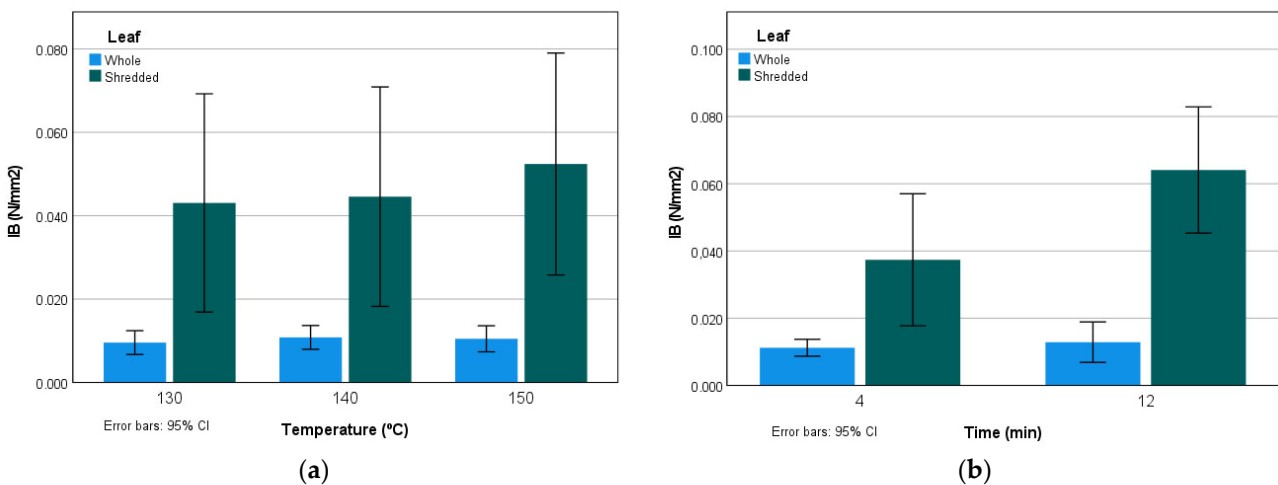

**Figure 9.** Internal bonding strength (IB) according to (**a**) temperature and (**b**) pressing time.

According to European Standards [56], the minimum requirements to be classified as P1 (boards for general use in dry conditions) are a MOR value of 10.5 N/mm$^2$ and an IB value of 0.28 N/mm$^2$. These experimental panels are still far from being used for these applications. However, they have dimensional stability and are water resistant; therefore, they could be used as insulation panels.

### 3.3. Thermal Properties

The thermal conductivity of the panels ranged from 0.065 to 0.085 W/m·K in whole-leaf boards and from 0.080 to 0.089 W/m·K in shredded-leaf boards (Figure 10). These are good results in comparison to commercial particle boards [57] with values of 0.180 W/m·K and similar to cork boards with 0.065 W/m·K. Figure 10 indicates that since the deviation bars overlap, no relationship could be found with the other two manufacturing variables.

Modern commercial insulation panels have thermal conductivities that range from 0.030 to 0.040 W/m·K but are manufactured with a high environmental cost. The olive pruning leaf panels could replace them, but it would be necessary to double the thickness of the panels, which, due to the density of the boards, is not operational. To compete with modern insulators, it would be compulsory to improve insulating capabilities or reduce densities.

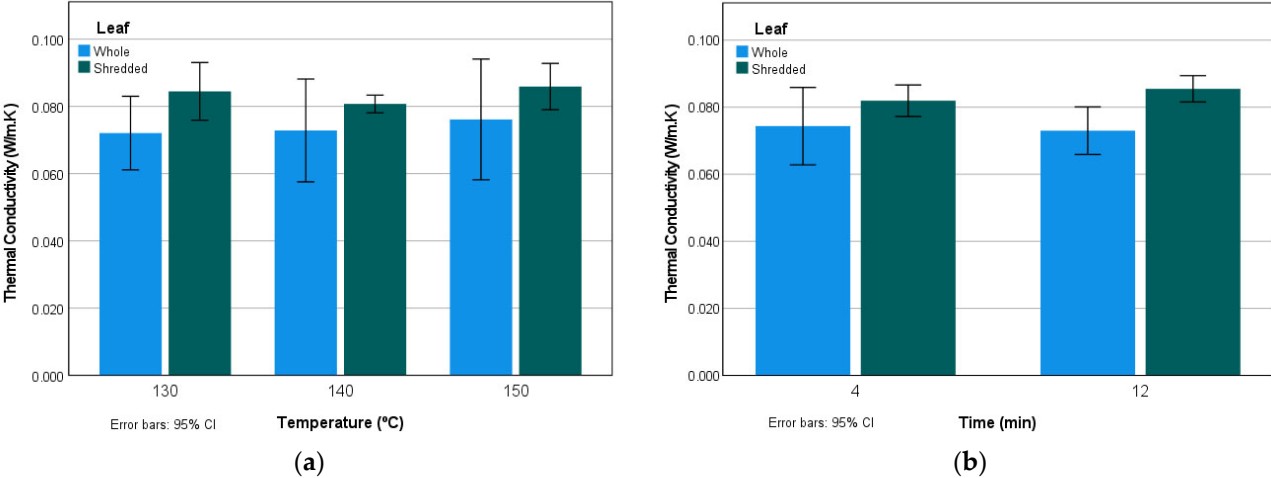

**Figure 10.** Thermal conductivity according to (**a**) temperature and (**b**) pressing time.

*3.4. Statistical Analysis*

As can be seen in Table 3, all the parameters have dependency relationships (sig < 0.05) with the type of leaf (whole leaves or shredded). The particle size of the shredded leaves ranged from 3 to 4 mm in diameter. It should be studied whether reducing this size would improve the properties of the panels.

**Table 3.** ANOVA of the results of the tests.

| Factor | Properties | Sum of Squares | d.f. | Half Quadratic | F | *p*-Value |
|---|---|---|---|---|---|---|
| Leaf type | Density (kg/m$^3$) | 84,534.65 | 1 | 84,534.65 | 10.99 | 0.001 |
| | TS 24 h (%) | 1551.53 | 1 | 1551.52 | 49.02 | <0.001 |
| | WA 24 h (%) | 1213.56 | 1 | 1213.56 | 8.12 | 0.006 |
| | MOR (N/mm$^2$) | 15,08 | 1 | 15.08 | 26.26 | <0.001 |
| | MOE (N/mm$^2$) | 754,422.53 | 1 | 754,422.53 | 15.24 | <0.001 |
| | IB (N/mm$^2$) | 0.03 | 1 | 0.03 | 31.39 | <0.001 |
| | Thermal C. (W/m·K) | 0.00 | 1 | 0.00 | 12.69 | 0.002 |
| Pressing temperature | Density (kg/m$^3$) | 93,802.05 | 2 | 46,901.03 | 6.10 | 0.003 |
| | TS 24 h (%) | 376.08 | 2 | 188.04 | 3.83 | 0.027 |
| | WA 24 h (%) | 255.16 | 2 | 127.58 | 0.77 | 0.467 |
| | MOR (N/mm$^2$) | 1.19 | 2 | 0.60 | 0.89 | 0.416 |
| | MOE (N/mm$^2$) | 799,835.17 | 2 | 399,917.59 | 8.07 | <0.001 |
| | IB (N/mm$^2$) | 0.00 | 2 | 0.00 | 0.13 | 0.879 |
| | Thermal C. (W/m·K) | 0.00 | 2 | 0.00 | 0.49 | 0.619 |
| Pressing time | Density (kg/m$^3$) | 115,661.14 | 1 | 115,661.14 | 15.47 | <0.001 |
| | TS 24 h (%) | 7.93 | 1 | 7.93 | 0.15 | 0.702 |
| | WA 24 h (%) | 38.76 | 1 | 38.76 | 8.12 | 0.631 |
| | MOR (N/mm$^2$) | 13.36 | 1 | 13.36 | 22.78 | <0.001 |
| | MOE (N/mm$^2$) | 130,082.85 | 1 | 130,082.85 | 15.24 | 0.123 |
| | IB (N/mm$^2$) | 0.00 | 1 | 0.00 | 3.04 | 0.085 |
| | Thermal C. (W/m·K) | 0.00 | 1 | 0.00 | 0.098 | 0.757 |

d.f.: degrees of freedom. F: Fisher–Snedecor distribution.

The properties that were dependent on pressing time were density and MOR. The size of the sample has not made it possible to assess the MOE and IB dependency relationship. Those that were dependent on pressing temperature were density, TS 24 h and MOE. As in the previous case, the WA 24 h was unrelated, and MOR and IB were as well.

According to Table 4, in order to improve the mechanical behavior of the panels (MOR, MOE and IB), a smaller particle size of shredded leaves is needed. This is especially influential in IB results. At the same time, the values of TS, WA and density would increase.

**Table 4.** Pearson correlation of the results of the tests.

| Factor | Pearson | Density (kg/m$^3$) | TS 24 h (%) | WA 24 h (%) | MOR (N/mm$^2$) | MOE (N/mm$^2$) | IB (N/mm$^2$) | Thermal. C. (W/m·K) |
|---|---|---|---|---|---|---|---|---|
| Leaf type | Correlation | −0.131 | −0.612 ** | −0.335 ** | −0.257 ** | −0.259 ** | −0.591 ** | −0.396 |
| | Sig (bilateral) | 0.118 | <0.001 | 0.004 | 0.002 | 0.002 | <0.001 | 0.055 |
| | N | 144 | 72 | 72 | 144 | 144 | 72 | 24 |
| Pressing temp. | Correlation | 0.231 ** | 0.277 * | 0.147 | 0.111 | 0.301 ** | 0.045 | 0.136 |
| | Sig (bilateral) | 0.005 | <0.019 | 0.218 | 0.187 | <0.001 | 0.707 | 0.528 |
| | N | 144 | 72 | 72 | 144 | 144 | 72 | 24 |
| Pressing time | Correlation | 0.313 ** | −0.046 | 0.058 | 0.372 ** | 0.129 | 0.204 | 0.067 |
| | Sig (bilateral) | <0.001 | 0.702 | 0.631 | <0.001 | 0.123 | 0.085 | 0.757 |
| | N | 144 | 72 | 72 | 144 | 144 | 72 | 24 |

* The correlation is significant at 0.05 level (bilateral). ** The correlation is significant at 0.01 level (bilateral).

If pressing temperature is higher, denser panels will be manufactured with more TS 24 h values and better MOE performance. With more pressing time, the panels will have more density and MOR values. It is important to note that density plays an important role in mechanical performance of the boards.

With less pressing temperature, pressing time and whole leaves, TS 24 h and density will decrease.

Contrary to ANOVA statistics, the thermal conductivity was not influenced by any of the manufacture values. This was probably due to the small number of tests of the boards, with only 24 in total. To improve the thermal insultation capacity of the panels, further research is needed.

*3.5. Comparison with Other Studies of Panels Made of Leaves*

Other studies have used leaves of different plants to manufacture panels (Table 5). Using typha leaves, Dieye et al. [58] achieved excellent conductivity results (0.055 to 0.083 W/m·K). However, their panels required a natural binder (gum arabic) in very large concentrations (from 33 to 50% wt). Tangjuank [59] studied the properties of thermal insulation produced from pineapple leaves using another natural rubber latex as a binder (in proportions from 1:2 to 1:4). He obtained a thermal conductivity that ranged from 0.035 to 0.043 W/m·K with densities from 178 to 210 kg/m$^3$. The author built a box composed of two panels and an air chamber inside, which improved its insulation capacity.

This study achieves binderless panels with thermal conductivity values from 0.065 to 0.089 W/m·K. These results could be improved by using natural binders or decreasing the density of the panels.

With different adhesives, other authors have manufactured panels with good mechanical and physical results. Masturi et al. [60] made a composite of northern red oak leaves with 4–20% wt polyurethane. They discovered that, starting with 12% of polyurethane (PU), a good WA 24 h was achieved. Yalinkilic et al. [61] made a tri-layer panel with waste tea leaves with 8 to 10% urea-formaldehyde (UF) wt and 1% paraffin wax. Some of the panels could be used in general applications. Aghakhani et al. [62] used sycamore leaves with 4% of Methylene Diphenyl Diisocyanate (MDD) as a binder. Their overall results showed very good values, which exceeded the minimum requirements of European Standards for furniture manufacturing.

Nemli et al. [63] manufactured grass clipping waste panels with 12% UF. Their panels showed low results in comparison to when they added eucalyptus wood particles for manufacturing the panels. They discovered that higher concentrations of grass decreased the performance of the panels and considered that this was probably caused by the higher amount of lignin and lower cellulose content of the grass clippings. A similar study researching the influence of adding particles of gingko tree leaves (1, 5 and 10%) to conventional wood with 10% UF [64] showed that the addition of a small amount of gingko leaves did not harm the physical and mechanical properties of particleboards made of wood.

**Table 5.** Properties obtained with boards made of leaves.

| Source | Material | Binder | Density (kg/m$^3$) | TS 24 h (%) | WA 24 h (%) | MOR (N/mm$^2$) | MOE (N/mm$^2$) | IB (N/mm$^2$) |
|---|---|---|---|---|---|---|---|---|
| [60] | Red oak | 4–20% PU | 825–1261 | | 1.38–107.35 | | | |
| [61] | Tea | 8–10% UF | 550–750 | 7.50–41.00 | 37.50–135.00 | | | 2.20–12.50 |
| [62] | Sycamore | 4% MDD | 600–700 | 15.28–18.05 | 31.70–34.20 | 15.92–17.23 | 1958–2130 | 0.58–0.74 |
| [63] | Grass | 12% UF | | 33.6 | | 4.19 | 351.35 | 0.08 |
| [48] | Canary palm | binderless | 855 | 54.32 | 132.53 | 6.26 | 1045.92 | 0.29 |
| [65] | Oil palm | binderless | 800 | 65.00 | 130.00 | 2.00 | | 0.00 |
| This work | Olive | binderless | 888–936 | 16.18–25.46 | 48.15–56.36 | 5.02–5.67 | 764.62–909.38 | 0.01–0.05 |

Two other investigations were focused on obtaining binderless panels of leaves from palm trees. Ferrandez-Garcia et al. [48] used canary palm rachis (part of the leaf) to manufacture binderless particleboards. The panels showed good mechanical properties but low physical results. Hashim et al. [65] used oil palm fiber leaves to manufacture binderless panels. They lacked internal bonding strength; hence, the rest of the mechanical properties were very low. Panels made of leaves of olive waste pruning have the potential to improve these results, and it should be stressed that the boards in this work offered better properties than those achieved with other leaves' fibers without adhesives.

## 4. Conclusions

Boards with olive pruning leaves without adhesives have been successfully obtained with good results of thickness swelling (TS) at 24 h, water absorption (WA) at 24 h and of thermal conductivity and can be used as an alternative for manufacturing thermal insulation boards.

With a smaller particle size of shredded-leaf panels, the mechanical behavior of the boards can improve (MOR, MOE and IB). With longer pressing times and higher pressing temperatures, density can increase, improving the mechanical values since it plays an important role in the mechanical performance of the boards. With less pressing temperature, pressing time and whole leaves, TS 24 h and density will decrease.

The olive leaves are a low-cost renewable resource, and manufacturing products with a long, useful life can be beneficial to the environment, as it is a method of carbon fixation and therefore contributes to reducing $CO_2$ in the atmosphere.

**Author Contributions:** Idea and methodology: A.F.-G. and M.T.F.-G. Experiments: M.T.F.-G. and T.G.O. Resources: T.G.O. Statistics: M.F.-V. and A.F.-G. Project administration: A.F.-G. and F.M.-C. Supervision: F.M.-C. and M.F.-V. Writing: A.F.-G. Review: T.G.O. All authors have read and agreed to the published version of the manuscript.

**Funding:** This research was funded thanks to Agreement No. 4/20 between the company Aitana, Actividades de Construcciones y Servicios, S.L. and Universidad Miguel Hernandez, Elche.

**Institutional Review Board Statement:** Not applicable.

**Informed Consent Statement:** Not applicable.

**Data Availability Statement:** The data presented in this study are available within the article.

**Acknowledgments:** The authors would like to thank the company Aitana, Actividades de Construcciones y Servicios, S.L. for its support by signing Agreement No. 4/20 with Universidad Miguel Hernández, Elche on 20 December 2019.

**Conflicts of Interest:** The authors declare no conflict of interest.

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
