# Peer review of "Analysis of the Manufacturing Variables of Binderless Panels Made of Leaves of Olive Tree (Olea europaea L.) Pruning Waste"

_agronomy, doi:10.3390/agronomy12010093_

Round 1

Reviewer 1 Report

This manuscript is well written and can raise the interest of a braod readership due to the environmental context. Waste valorization is an important concern and construction panel is one valorization way that should be considered. Herein the authors propose a binder-free methodology to process boards. The results are correctly presented with statistical treatment to extract most significant information. I have few comments to be considered by the authors before publication.

Line 62: could the authors add whether the previously cited works include binders in the formulation or not?

Line 86: Please add a reference

Table1: Please define N

Experimental part: Please provide the number of tests replica

On Figures: Please mention at least one time that CI is confidence interval

Line 147: What is written is true. On the other hand, shredded leaves absorb more. This might influence TS.

Paragraph 3.2: MOR in the order of 5 MPa, MOE in the order of GPa. We can range this as medium resistant and medium rigid. Could the authors provide an idea about typical expectations for these boards? 

Table2: legend: Sig. does not correspond to any column

Line 210 and Table 3: This seems to unmatch. Table 3 gives Pearson analysis. However nothing regarding particle size is mentioned.

Line 246: Please replace by "Their panels 245 showed low results. This is probably caused". Please detail what means "low results"

Reviewer 2 Report

The main subject of the authors' research was the use of the original method to valorize the leaves of olive tree pruning waste consisting of the manufacture of ecologic boards without adhesives. The primary materials used to manufacture panels were leaves from olive tree pruning and water from the municipal network. The authors described the method of production, physical and mechanical properties of new boards. The work has enormous innovative and inventive potential from a practical point of view and has a high cognitive potential scientifically.

Detailed comments: 

  1. How was the size of chips of shredded material measured? What was the particle size after shredding compared to other known materials, e.g. particleboard?
  2. Did the pressing result in 24 types of plates of the same thickness?
  3. If the pressure was maintained at 2.1 MPa, so is it possible that whole leaves are less compressed on the hot press, and the panels have less density? Maybe it is due to the amount of substances released during ironing?
  4. The research results clearly showed an increase in MOR and MOE of boards pressed at a temperature of 150 degrees. However, such a significant increase in density was not observed. Therefore, the authors should explain the reasons for panels' increasing mechanical properties at this temperature.
  5. Statistical analysis is sufficiently correct when analyzing the influence of one factor on many features. In my opinion, however, the MANOVA analysis would be more favorable.
  6. I consider comparing the results with the works of other authors as a successful part of the work. The discussion included here has a high scientific value.
